# Ligand’s Partition to the Lipid Bilayer Should Be Accounted for When Estimating Their Affinity to Proteins

**DOI:** 10.3390/molecules28073136

**Published:** 2023-03-31

**Authors:** Maria João Moreno, Armindo Salvador

**Affiliations:** 1Department of Chemistry, Coimbra Chemistry Center, Institute of Molecular Sciences (CQC-IMS), University of Coimbra, 3004-535 Coimbra, Portugal; 2CNC—Center for Neuroscience and Cell Biology, University of Coimbra, 3004-504 Coimbra, Portugal; 3Institute for Interdisciplinary Research, University of Coimbra, 3030-789 Coimbra, Portugal

**Keywords:** binding affinity, partition coefficient, membrane proteins, lipid-protein ratio, ligand sequestration, ligand exclusion, protein specificity

## Abstract

Ligand-protein interactions are usually studied in complex media that also contain lipids. This is particularly relevant for membrane proteins that are always associated with lipid bilayers, but also for water-soluble proteins studied in in vivo conditions. This work addresses the following two questions: (i) How does the neglect of the lipid bilayer influence the apparent ligand-protein affinity? (ii) How can the intrinsic ligand-protein affinity be obtained? Here we present a framework to quantitatively characterize ligand-protein interactions in complex media for proteins with a single binding site. The apparent affinity obtained when following some often-used approximations is also explored, to establish these approximations’ validity limits and to allow the estimation of the true affinities from data reported in literature. It is found that an increase in the ligand lipophilicity or in the volume of the lipid bilayer always leads to a decrease in the apparent ligand-protein affinity, both for water-soluble and for membrane proteins. The only exceptions are very polar ligands (excluded from the lipid bilayer) and ligands whose binding affinity to the protein increases supralinearly with ligand lipophilicity. Finally, this work discusses which are the most relevant parameters to consider when exploring the specificity of membrane proteins.

## 1. Introduction

The interaction of ligands with proteins governs most biological processes, from the interaction of substrates, inhibitors, and modulators with enzymes, to the interaction of hormones with receptors. The amount of ligand that binds to the protein is influenced by the presence of the lipid membrane, both for proteins associated with biomembranes and for proteins soluble in the aqueous media. On one hand, the association of the ligand with the membrane increases the local concentration in the environment where a membrane protein is located, expectably facilitating binding to the protein. On the other hand, for water-soluble proteins, the association of the ligand with the membrane leads to a decrease in its concentration in the aqueous medium, decreasing the amount of ligand that may bind to the protein [1]. However, these effects are often overlooked in analyses of the affinity of hydrophobic ligands to proteins, leading to erroneous conclusions. Here we analyze the conditions where membrane association cannot be neglected, and we present a method to take it into account. The properties of membrane proteins are also influenced by the lipid bilayer where they are embedded [2,3,4,5,6,7,8]. In spite of its importance, this aspect will not be considered in this work. The lipid bilayer will be considered only in what concerns ligand association, overlooking any direct effect on the properties of the protein.

The effects of the lipid bilayer on the kinetics of ligand binding to membrane proteins has been extensively explored. In this case, a faster association is expected, due to the increase in the local ligand concentration, the well-defined orientation of the ligand and the two-dimensional approach of the ligand towards the protein [9,10,11]. However, a quantitative analysis of how the presence of the lipid bilayer influences the equilibrium association of ligands to proteins has not been performed. An important distinction in this respect is between the intrinsic binding affinity (which reflects the interactions between the ligand and the protein), and the apparent affinity obtained experimentally at specific assay conditions. Due to the unavailability of an appropriate formalism to quantitatively describe the ligand-protein association in complex media, the binding affinity is usually obtained assuming that all ligand is available. This leads to apparent binding affinities that are dependent on the specific conditions of the assay.

P-glycoprotein (P-gp) is an example of an intrinsic membrane protein, and the protein binding pocket is usually considered accessible by the ligand in the lipid portion of the membrane [12,13,14]. Therefore, for this protein, an increase in the ligand lipophilicity leads to an increase in the local concentration in the membrane, and an increase in the apparent affinity is usually observed [15,16]. On the other hand, for very lipophilic ligands, increasing the volume of the lipid phase may lead to a dilution of the ligand in the membrane, thus resulting in a decrease in the apparent affinity [17].

If the protein is in the aqueous medium, or associated with the membrane but with its binding pocket only accessible to the ligand in the aqueous phase, the association of the ligand in the lipid phase is expected to lead to a decrease in the apparent affinity obtained from the ligand’s overall concentration. When comparing two ligands with the same intrinsic affinity but different lipophilicity, a stronger decrease is expected for the more lipophilic ligand, incorrectly suggesting that lipophilicity has a negative contribution to protein specificity.

When studying the affinity of a given ligand to a specific protein at the same lipid-to-protein ratio but under different conditions, factors affecting the ligand’s interaction with the lipid membrane may also influence the apparent affinities. This may occur, for example, when the membrane lipid composition varies over those conditions. Then, because the partition of small molecules to lipid bilayers is strongly affected by the lipid composition [18,19,20,21,22,23,24,25,26,27,28], the distribution of the ligand between the distinct media changes, leading to different apparent affinities for the protein. Another important example occurs when comparing studies performed at distinct pH values. Because most ligands have weak acid/base groups [29], their ionization state depends on pH. The ligands’ affinity for the lipid bilayer depends on their global charge and location of the ionized groups. Therefore, the change in pH alters the amount of ligand associated with the membrane [20,24,30,31,32,33,34,35]. So, the effect of pH on the ligand-protein apparent affinity will reflect not only the specificities of ligand-protein interaction, but also its distinct affinity for the lipid bilayer.

The formalisms to quantitatively characterize the equilibrium distribution of ligands in media containing proteins and lipid membranes have been recently reviewed by us [1]. Focus was given to the importance of using the local concentration of ligand in each medium, instead of the usually considered overall ligand concentration. This formalism has been applied to the characterization of the interaction of a homologous series of ligands with P-glycoprotein [15] and allowed the characterization of the ligands’ intrinsic affinities. Importantly, the same intrinsic affinities were obtained when the interaction was characterized by two distinct assays (ATPase activity and ligand displacement), performed at different membrane concentrations (0.1 and 1 mg protein/mL, respectively).

The examples above show that changes in the apparent affinity cannot be used to evaluate protein specificity. Apparent affinities include contributions from variations in the ligand affinity towards both the lipid membrane and the protein. Moreover, they depend on the conditions of the assay, such as the amount of membrane and the lipid-protein ratio. Those limitations have impeded the elucidation of ligand specificity for membrane proteins. As such, they contribute to the high attrition rates in drug development, with a poor efficacy in vivo for active principles optimized from in vitro studies based on apparent affinities and activities.

For an accurate and quantitative analysis of ligand-protein interactions in the presence of lipid membranes, the partition of the ligand to the membrane must be known and included in the analysis [15,36]. It will be shown in this article that the quantitative evaluation of the ligand interaction with the lipid portion of the membrane is necessary for the quantitative evaluation of the affinity of the ligand to the protein. The characterization of the interaction of the ligand with the specific membrane is therefore of major importance and should not be simply estimated from its partition between water and octanol (see, e.g., [15,37,38,39,40,41,42] for discussions on the limitations of this approach).

This work presents a quantitative evaluation of the effect of the amount of lipid membrane and of the lipid-protein ratio on the apparent ligand-protein affinities, when evaluated from the overall ligand concentration. Several cases will be considered, from proteins in solution in the presence of lipid membranes to biomembranes containing lipids and intrinsic membrane proteins. The objective is to call the attention of the scientific community to the importance of including the lipid membrane in the analysis of the ligand-protein interaction and to provide the framework for an accurate characterization of the intrinsic ligand-protein affinity.

## 2. Results

As indicated in reference [1], the distribution of the ligand between the distinct media (aqueous, protein and lipid membrane) may be described by a partition coefficient towards the membrane and a binding association with the protein; alternatively, partition coefficients may be considered for the association of the ligand with both binding agents. The former analysis is preferable when the molar concentration of the protein is not well known (only its total mass is known, for example) or when the ligand occupancy number in the protein is not well defined (as is the case for proteins with large binding pockets corresponding to several possibly overlapping binding sites). The comparison between the two formalisms may provide information on the number of ligands that can bind to the protein binding pocket, as was recently carried out for the interaction of a series of ligands to P-glycoprotein [15].

The schemes for the equilibrium of the ligand association with the protein and the lipid membrane are presented in Figure 1 for the case of water-soluble proteins (case I) and membrane proteins (case II). For simplicity, it is considered that the protein contains a single binding site. It is not defined a priori whether the ligand binds to the protein from the aqueous phase or from the lipid bilayer. In fact, the three equilibria form a thermodynamic cycle, and therefore the equilibrium constants are related through the micro-reversibility constraint presented below. Thus, even if binding to the protein occurs only from one of the phases (aqueous or lipid membrane), the equilibrium constant for binding from the other medium is not independent. To establish the quantitative relation between all three equilibrium constants, it is necessary to express them in terms of a partition coefficient, Equation (1) [1,15].
(1)$$K_{\mathrm{eq}}^{L_{X}}=K_{P_{P}}^{L_{X}}\bar{V_{P}}$$

Here, $\bar{V_{P}}$ is the molar volume of the protein, X is the medium (aqueous, W, or the lipid bilayer, Lb) from which the ligand binds the protein, $K_{P_{P}}^{L_{x}}$ is the ligand’s partition coefficient between medium X and the protein, and $K_{\mathrm{eq}}^{L_{x}}$ is the binding equilibrium constant. The relation between the three partition coefficients is given by Equation (2).
(2)$$K_{P_{P}}^{L_{W}}=K_{P_{\mathrm{Lb}}}^{L_{W}}\;K_{P_{P}}^{L_{\mathrm{Lb}}}$$

The previous two equations are only valid for diluted media, where the protein occupies a much smaller volume than that of the medium with which the ligand equilibrates. The general equations and their derivation are provided in Appendix A (Section A.1 and Section A.2).

For water-soluble proteins, it is possible to characterize $K_{P_{P}}^{L_{w}}$ (or the corresponding $K_{P_{P}}^{L_{w}}$) in the absence of the lipid membrane and $K_{P_{\mathrm{Lb}}}^{L_{w}}$ in the absence of the protein. From those two equilibria, the third ($K_{P_{P}}^{L_{\mathrm{Lb}}}$) may be calculated using Equation (2), even if it does not correspond to an observed path of ligand distribution. On the other hand, for membrane proteins, it is not possible to directly characterize $K_{\mathrm{eq}}^{L_{W}}$ (nor the corresponding $K_{P_{P}}^{L_{w}}$) because the membrane must always be present. In this case, the two equilibrium constants that can be directly obtained are $K_{P_{\mathrm{Lb}}}^{L_{W}}$ (in the presence of an equivalent lipid membrane but without the protein) and the overall affinity for the membrane containing the protein $K_{\mathrm{PM}}^{L_{W}}$, which is related with the other equilibria by Equation (3),
(3)$$K_{P_{M}}^{L_{W}}=K_{P_{\mathrm{Lb}}}^{L_{W}}\frac{V_{\mathrm{Lb}}}{V_{M}}+K_{P_{P}}^{L_{W}}\frac{V_{P}}{V_{M}}\;$$
with the volume of the membrane ($V_{M}$) being equal to the sum of the volume of the lipid bilayer ($V_{\mathrm{Lb}}$) *plus* the volume of the protein ($V_{P}$) in the membrane. See Section A.3 for derivation. Note that in this case, the volume occupied by the protein cannot be neglected, even if the protein is diluted. This is because the volume is multiplied by the partition coefficient, with a significant contribution even if the volume of the protein is much smaller than that of the lipid bilayer.

Knowing $K_{P_{\mathrm{Lb}}}^{L_{W}}$, $K_{P_{M}}^{L_{W}}$, and the amount of protein and lipid bilayer, it is thus possible to obtain $K_{P_{M}}^{L_{W}}$, and the ligand equilibrium between the lipid bilayer and the protein ($K_{P_{P}}^{L_{\mathrm{Lb}}}$) may be calculated using Equation (2).

Due to the thermodynamic cycle that connects the ligand between all the distinct media, the equations that describe the equilibrium distribution of ligand are the same, regardless of the protein being soluble in the aqueous phase or a membrane protein, and whether the ligand binds to the protein from the aqueous medium or from the lipid bilayer. The only difference is in which of the two equilibria can be directly characterized experimentally.

In this section, the results from kinetic modeling considering the ligand-protein and ligand–membrane interactions will be provided for the case of proteins soluble in the aqueous phase (Section 2.1) and membrane proteins (Section 2.2.1). Some specific examples from literature for the membrane protein P-gp are presented and discussed in Section 2.2.2, and some considerations regarding the evaluation of the specificity of membrane proteins will be made in Section 2.2.3.

### 2.1. Proteins Soluble in the Aqueous Phase

For the interaction of a ligand with a protein soluble in the aqueous phase in the presence of a lipid membrane, the two equilibria that may be directly characterized, and the corresponding equations, are:(4)$$L_{W}\overset{K_{P_{\mathrm{Lb}}}^{L_{W}}}{↔}L_{\mathrm{Lb}}\;;\;\left[L_{\mathrm{Lb}}\right]=\left[L_{W}\right]K_{P_{\mathrm{Lb}}}^{L_{W}}\frac{V_{\mathrm{Lb}}}{V_{W}}\;L_{W}\overset{K_{P_{\mathrm{Lb}}}^{L_{W}}}{↔}L_{\mathrm{Lb}}\;;\;\left[L_{\mathrm{Lb}}\right]=\left[L_{W}\right]K_{P_{\mathrm{Lb}}}^{L_{W}}\frac{V_{\mathrm{Lb}}}{V_{W}}\;\left[L_{T}\right]=\left[L_{W}\right]+\left[L_{\mathrm{Lb}}\right]+\left[L_{P}\right]\;\left[P_{T}\right]=\left[P\right]+\left[L_{P}\right]$$
where the concentrations are all with respect to the total volume of the solution. As indicated before, those equations are only valid for diluted solutions. The corresponding equations for any amount of protein and/or lipid bilayer are provided in Section A.4.

The concentration of ligand free in the aqueous medium is obtained from the set of equations above, leading to a quadratic equation that may be solved analytically, Equation (5),
(5)$${\left[L_{W}\right]}^{2}K_{\mathrm{eq}}^{L_{W}}\left(1+K_{P_{\mathrm{Lb}}}^{L_{W}}\frac{V_{\mathrm{Lb}}}{V_{W}}\right)+\left[L_{W}\right]\left(1-K_{\mathrm{eq}}^{L_{W}}\left(\left[L_{T}\right]-\left[P_{T}\right]\right)+K_{P_{\mathrm{Lb}}}^{L_{W}}\frac{V_{\mathrm{Lb}}}{V_{W}}\right)-\left[L_{T}\right]=0$$
with the solution of the quadratic equation, which has physical meaning being always *x*_+_ [1]. The concentration of ligand bound to the protein and associated with the lipid bilayer is calculated from $\left[L_{W}\right]$ using Equation (4).

If the protein contains several binding sites, the concentration of free ligand may have to be obtained numerically [43,44,45,46,47,48,49,50,51].

The protein saturation predicted for a moderate affinity of ligand binding to both the protein and the lipid bilayer ($K_{\mathrm{eq}}^{L_{W}}$ = 10^6^ M^−1^, and $K_{P_{\mathrm{Lb}}}^{L_{W}}$ = 10^3^), a medium-size protein (M_W_= 50 kDa) at a total concentration of 10 μM, and different amounts of a lipid bilayer, is shown in Figure 2.

As expected, the presence of the lipid bilayer leads to the sequestration of ligand and decreases the amount of ligand in the aqueous medium (see Appendix B, Figure A1) and associated with the protein. If protein saturation is described with a model that neglects ligand association with the lipid bilayer, Equation (6), the binding affinity obtained is apparent ($K_{\mathrm{eqApp}}^{L_{W}}$), is lower than the effective one, and depends on the amount of lipid bilayer present. The quality of the best fit is always excellent, nevertheless (Figure 2A lower plot).
(6)$$\begin{array}{l} {\left[L_{W}\right]}^{2}K_{\mathrm{eqApp}}^{L_{W}}+\left[L_{W}\right]\left(1-K_{\mathrm{eqApp}}^{L_{W}}\left(\left[L_{T}\right]-\left[P_{T}\right]\right)\right)-\left[L_{T}\right]=0\;\mathrm{solution}\;x_{+}\\ \left[L_{P}\right]=\left[L_{T}\right]-\left[L_{W}\right]\\ \left[P\right]=\left[L_{T}\right]-\left[L_{P}\right] \end{array}$$

If, incorrectly but unfortunately very frequent, the protein saturation is analyzed with a model that considers that the ligand is in large excess relative to the protein, Equation (7), the best fit is poor (Figure 2B bottom plot).
(7)$$\begin{array}{ccc} \left[L_{W}\right] & ≅ & \left[L_{T}\right]\\ \left[P\right] & = & \left[L_{T}\right]-\left[L_{P}\right]\\ \left[L_{P}\right] & = & \frac{\left[L_{T}\right]K_{\mathrm{eqApp}}^{L_{T}}\left[P_{T}\right]}{1+K_{\mathrm{eqApp}}^{L_{T}}\left[P_{T}\right]} \end{array}$$

The values obtained for the apparent ligand affinity are represented in Figure 3, for the total concentration of protein considered in Figure 2 (10 μM) and for a lower (1 μM) or higher (100 μM) concentration of protein. In plot A, the apparent affinity when analyzing the data with Equation (6) is represented, assuming that the ligand is either in the aqueous medium or associated with the protein, $K_{\mathrm{eqApp}}^{L_{W}}$. While in plot B, it is the apparent affinity when assuming a large excess of ligand, $K_{\mathrm{eqApp}}^{\mathrm{LT}}$ Equation (7).

When sequestration of the ligand by the lipid bilayer is ignored, and the ligand binding to the protein is analyzed with Equations (6), the apparent binding affinity deviates from the true affinity, getting smaller as the amount of lipid bilayer increases (Figure 3A). The inaccuracy in the binding affinity is independent of the concentration of protein and is described by Equation (8),
(8)$$\frac{K_{\mathrm{eqApp}}^{L_{W}}}{K_{\mathrm{eq}}^{L_{W}}}=\frac{1}{1+K_{P_{\mathrm{Lb}}}^{L_{W}}\alpha_{\mathrm{Lb}}}$$

This equation is valid in dilute solutions (α_W_ ≥ 0.99); see Section A.5 for its derivation and for the corresponding equation in concentrated media.

If both the ligand associated with the lipid bilayer and that bound to the protein are neglected, Equation (7) Figure 3B, the inaccuracy in the estimated binding affinity is even larger and depends on the total concentration of protein. As expected, the larger the amount of lipid bilayer and the concentration of protein, the smaller the estimated apparent binding affinity.

The effect of ligand sequestration by a lipid phase on the apparent binding affinity of the ligand to the protein is of relevance when the system in study corresponds to an extract from a biological sample without extensive purification. However, in most studies of ligand-protein binding a purified protein is used, with little or no contamination from lipids. The above is very relevant, when whole cells or organisms are being studied and when the in vivo binding affinity is predicted from studies in vitro using purified samples. In this case, the use of the binding affinity obtained in vitro and neglecting ligand association to the lipidic phases in vivo will overestimate the amount of ligand bound to the protein in the complex system. For a correct description of the ligand distribution in a complex system, it is necessary to consider the intrinsic affinities and to use a formalism that explicitly includes the different binding agents at the concentration observed in the specific system.

Serum albumin is an example of a protein that binds ligands with moderate-to-high lipophilicity and where the amount of ligand bound in vivo may be significantly different from that predicted from the binding affinity obtained in vitro with purified protein. Blood plasma contains about 600 μM albumin (corresponding to *V*_P_/*V*_T_ ≈ 3%), [52] and a high number of lipoproteins that contain a lipophilic core of neutral lipids stabilized by a phospholipid layer and proteins, corresponding to *V*_Lb_/*V*_T_ ≈ 1%, [53,54,55,56,57]. Lipophilic drugs will partition towards the lipoproteins, which will decrease the apparent binding affinities to the serum albumin in the plasma. As an example, the drug chlorpromazine binds to albumin with moderate affinity ($K_{\mathrm{eq}}^{L_{W}}$ ≈ 10^6^ M^−1^ [52,58]) and presents a relatively high partition coefficient to lipid bilayers ($K_{\mathrm{PLb}}^{L_{W}}$ ≈ 10^4^ [21,59,60]). In the plasma, it is predicted from Equation (8) that the apparent binding affinity of chlorpromazine to serum albumin is decreased to ≈10^4^ M^−1^, with about half of the drug being associated with the lipidic phase in the lipoproteins. The apparent affinity of the drug to serum albumin is further decreased when the whole blood is considered due to the erythrocytes’ membranes (leading to *V*_Lb_/*V*_T_ ≈ 1.5%) [57,61,62] and, in particular, in the capillaries, due to the membrane of the endothelial cells (which increase *V*_Lb_/*V*_T_ to somewhat above 1.5%). It is therefore essential to include both the proteins and the lipid phases in the formalisms used to describe and predict drug pharmacokinetics from the binding affinities obtained in purified model systems [42,63,64,65].

### 2.2. Membrane Proteins

Most proteins in the cell interact with biomembranes at some point, and this is important for their biological function. About one third of the genes encode for intrinsic membrane proteins [66,67], which interact strongly with the lipid bilayer and are an inherent component of the membrane. However, proteins soluble in the aqueous medium may also associate with the cell membranes, transiently or more permanently, through interactions with the lipid bilayer or with membrane proteins [68,69,70,71,72,73]. The presence of the lipid bilayer influences their function, not only due to direct interactions [2,3,4,5,6,7,8], but also indirectly, through interactions with their ligands [9,10,11,12,15]. The major conceptual difference regarding the effect of the lipid bilayer on the apparent affinity of the ligand to the protein is whether it is the ligand in the aqueous phase or in the lipid bilayer that binds to the protein. Intuitively, it is anticipated that a decrease in the apparent affinity is observed when binding is from the aqueous phase (ligand sequestration) and an increase if binding is from the lipid bilayer (increase in the local concentration). However, the lack of an appropriate formalism has impeded the quantitative analysis of the lipid bilayer effects. Instead, the ligand binding is usually characterized as an apparent affinity, without explicitly considering the ligand associated with the lipid bilayer. Moreover, in most situations, ligand-protein binding is analyzed assuming excess ligand, thus neglecting *both* the ligand associated with the lipid bilayer and with the protein itself (e.g., [16,74,75,76,77,78]). In any case, the apparent affinity obtained depends on the affinity of the ligand to the lipid bilayer, which has been a major problem in the identification of the ligand specificity of membrane proteins. This limitation has an enormous impact on drug discovery, since membrane proteins are important drug targets [79,80,81,82,83].

The distinction between proteins that bind the ligand from the aqueous medium and those that bind the ligand from the lipid bilayer is illustrated in Figure 4. Regardless of the path followed by the ligand in the aqueous medium to bind to the protein (directly, case IIa; or mediated by the lipid bilayer, case IIb), the three equilibria form a thermodynamic cycle, and an equilibrium constant may be defined even for the path that cannot be followed by the ligand, Equation (2).

The two situations depicted in Figure 4 are therefore formally equivalent. They are also equivalent to the case of water-soluble proteins. Thus, the equations that should be used to quantitatively follow the equilibrium distribution of the ligand are those defined in Section 2.1.

There is, however, a major practical difference: it is not possible to characterize the binding affinity in the absence of the lipid bilayer. Thus, the intrinsic binding affinity cannot be directly characterized. Instead, it must be obtained from the amount of ligand bound to the protein in the presence of the lipid bilayer.

From the possible equilibria in the ligand association with the protein and lipid bilayer (Figure 1 and Figure 4), only $K_{P_{\mathrm{Lb}}}^{L_{W}}$ and $K_{\mathrm{PM}}^{L_{W}}$ may be directly characterized experimentally in the case of membrane proteins. The characterization of $K_{\mathrm{PP}}^{L_{W}}$ (and the corresponding intrinsic affinity, $K_{\mathrm{eq}}^{L_{W}}$) is therefore formalism-dependent. There are several alternative approaches to experimentally obtain $K_{\mathrm{eq}}^{L_{W}}$. The partition coefficient towards the whole membrane ($K_{\mathrm{PM}}^{L_{W}}$) and towards the lipid bilayer ($K_{P_{\mathrm{Lb}}}^{L_{W}}$) may be obtained at low ligand concentrations, allowing the calculation of $K_{P_{P}}^{L_{W}}$ (Equation (3)), and $K_{\mathrm{eq}}^{L_{W}}$ (Equation (1)). An alternative approach is to characterize the apparent binding affinity ($K_{\mathrm{eqApp}}^{L_{W}}$) at different volumes of the lipid phase and extrapolate to $V_{\mathrm{Lb}}$ = 0 using Equation (8). Finally, making use of this same equation, it is possible to calculate $K_{\mathrm{eq}}^{L_{W}}$ from the $K_{\mathrm{eqApp}}^{L_{W}}$ at a specific value *V*_Lb_, provided that the partition coefficient towards the lipid bilayer and its volume are known.

In Section 2.2.1, the formalism presented in Section 2.1 will be used to explore quantitatively the effect of the lipid bilayer on the distribution of ligand between the distinct media and on the apparent binding affinity obtained if the lipid bilayer is not taken into account. Then, in Section 2.2.2, we will present some specific cases taken from the literature and estimate the bias introduced by using the apparent affinity. Finally, Section 2.2.3 discusses some considerations into how protein specificity can be defined based on the binding affinities of a set of ligands with distinct properties.

#### 2.2.1. Effect of the Volume of the Lipid Bilayer and Ligand’s Lipophilicity on Ligand Binding to Membrane Proteins

When considering membrane proteins, it is important to distinguish whether they bind the ligand from the aqueous medium or from the lipid bilayer. Because the formalism for the analysis of ligand-protein binding is equivalent in both situations, the distinction can only be made in terms of ligand lipophilicity. Membrane proteins that bind very polar ligands must have their binding site accessible from the aqueous medium, while, for an efficient binding of lipophilic ligands, the binding site must be accessible from the lipid bilayer. The two situations will therefore be treated together, the distinction being made only in terms of ligand lipophilicity.

The effect of the volume of the lipid bilayer on the binding of ligands with distinct lipophilicity is represented in Figure 5. It should be noted that in what follows, the change in ligand lipophilicity may be due either to a different ligand being considered or to changes in the properties of the lipid bilayer or of the ligand. The latter may be due to a different lipid composition of the membrane or due to changes in the ligand’s ionization state in response to a distinct pH in the aqueous medium. The variation of the apparent binding constant between the aqueous phase and the protein $K_{\mathrm{eqApp}}^{L_{W}}$/$K_{\mathrm{eq}}^{L_{W}}$ is shown in the plots at the left. The variation of the fraction of ligand in the aqueous phase ([*L*_W_]/[*L*_T_], continuous lines) and associated with the lipid bilayer ([*L*_Lb_]/[*L*_T_], dashed lines) is shown on the middle plots. The protein saturation [*L*_P_]/[*L*_T_] is shown in the right plots. The intrinsic affinity for the protein was kept constant, $K_{\mathrm{eq}}^{L_{W}}$ = 10^6^ M^−1^, as well as the total concentration of protein, [*P*_T_] = 1 μM, while the volume of the lipid bilayer was increased from 0 to 15%.

The case of very polar ligands is represented in plots A to C. For ligands with equal affinity for the aqueous phase and the lipid bilayer ($Log\left(K_{P_{\mathrm{Lb}}}^{L_{W}}\right)$ = 0, red lines), the presence of the lipid bilayer does not affect the apparent binding affinity of the ligand to the protein, even for very high volumes of lipid bilayer. The fraction of ligand associated with the lipid bilayer increases and is accompanied by a decrease in the fraction of ligand in the aqueous phase, but the protein saturation with ligand remains essentially unchanged up to *V*_Lb_ = 15% (*v*/*v*). Surprisingly, at first, an increase in *V*_Lb_ leads to an increase in the apparent affinity of the ligand to the protein in the case of very polar ligands ($Log\left(K_{P_{\mathrm{Lb}}}^{L_{W}}\right)$ = −1, upper plots blue lines). This is because the ligand is excluded from the lipid bilayer, leading to an increase in its amount in the aqueous phase and associated with the protein. However, the effect is small, with an increase of 10% in the apparent affinity for 10% *v*/*v* of lipid bilayer.

The effect of the lipid bilayer in the distribution of ligands with moderate lipophilicity is shown in plots E to F ($Log\left(K_{P_{\mathrm{Lb}}}^{L_{W}}\right)$ = 2, blue lines; and $Log\left(K_{P_{\mathrm{Lb}}}^{L_{W}}\right)$ = 3, red lines). A strong decrease is observed in the apparent affinity for the protein, even for relatively low volumes of the lipid bilayer, the effect being stronger as the ligand lipophilicity increases. This is due to the sequestration of the ligand in the lipid bilayer, leading to a decrease in the ligand in the aqueous phase and associated with the protein. The same effect but more significant is observed for the case of ligands with high lipophilicity (plots G to I, $Log\left(K_{P_{\mathrm{Lb}}}^{L_{W}}\right)$ = 4, blue lines; and $Log\left(K_{P_{\mathrm{Lb}}}^{L_{W}}\right)$ = 5, red lines). In this case, the apparent affinity is decreased by more than an order of magnitude, for a volume of lipid bilayer equal to 0.1%.

The simulations shown in Figure 5 clearly show that to obtain the affinity of ligands to membrane proteins, the presence of the lipid bilayer must be accounted for in the analysis of the protein saturation with ligand. If the data is analyzed ignoring the association of the ligand with the lipid bilayer (as is common), the obtained apparent binding affinity does not reflect the strength of the interactions between the ligand and the protein, precluding the characterization of protein specificity. Another important drawback is that the decrease of $K_{\mathrm{eqApp}}^{L_{W}}$ with the volume of the lipid bilayer is dependent on ligand lipophilicity. Thus, the apparent affinity obtained for different lipid bilayer volumes and/or lipid-protein ratios cannot be compared. In contrast, if the intrinsic affinity is obtained, the amount of ligand bound at different volumes of lipid bilayer may be easily calculated.

In the simulations shown in Figure 5, it was assumed that the intrinsic affinity of the ligand to the protein is not affected by ligand lipophilicity. That is, the same value of $K_{\mathrm{eq}}^{L_{W}}$ was considered for all ligands, independently of their $K_{P_{\mathrm{Lb}}}^{L_{W}}$. Because the three equilibria are connected, the increase in $K_{P_{\mathrm{Lb}}}^{L_{W}}$ was accompanied by a decrease in $K_{P_{P}}^{\mathrm{LLb}}$ (Equation (2) and Figure 6A). As discussed above with respect to Figure 5, the apparent affinity of the ligand for the protein, $K_{\mathrm{eqApp}}^{L_{W}}$, is not significantly affected for ligands with low lipophilicity and decreases as the ligand lipophilicity increases. This apparent affinity was obtained from Equation (6), which neglect the ligand associated with the lipid bilayer and thus consider that all ligand not bound to the protein is in the aqueous phase. Figure 6 does not show the apparent affinity obtained when protein saturation is described by a model assuming excess ligand, $K_{\mathrm{eqApp}}^{\mathrm{LT}}$ Equation (7). This parameter would not only depend on the ligand lipophilicity but also on the concentration of protein, as shown in Figure 3 for the case of water-soluble proteins. However, it is not too much to repeat that, unfortunately, this is the formalism most commonly used to analyze the association of ligands with proteins, leading to apparent affinities that strongly depend on the ligand properties and on the parameters of the system considered in each assay.

Figure 6B represents the case in which the increase in the ligand lipophilicity leads to a proportional increase in the ligand affinity for the protein. In this case, $K_{\mathrm{Pp}}^{L_{\mathrm{Lb}}}$ is unchanged, and $K_{\mathrm{eq}}^{L_{W}}$ (and the corresponding $K_{\mathrm{Pp}}^{L_{W}}$) increases linearly with $K_{\mathrm{PLb}}^{L_{W}}$. The apparent binding affinity, $K_{\mathrm{eqApp}}^{L_{W}}$, also increases with $K_{\mathrm{PLb}}^{L_{W}}$. The increase is linear for ligands with low lipophilicity because the fraction of ligand associated with the lipid bilayer is negligible (at the low *V*_Lb_ considered in the simulations shown in Figure 6, 0.1% of the total volume, less than 1% of the ligand is associated with the lipid bilayer for $K_{\mathrm{PLb}}^{L_{W}}$ < 10). However, for ligands with moderate and high lipophilicity, sequestration in the lipid bilayer is significant, leading to a significant deviation from the intrinsic ligand affinity.

The extreme situation, where an increase in ligand hydrophobicity favors the binding to the protein more strongly than the binding to the lipid bilayer, is shown in Figure 6C. In this case, it was considered that $K_{\mathrm{PP}}^{L_{\mathrm{Lb}}}$ increases linearly with the ligand lipophilicity, leading to $K_{\mathrm{PP}}^{L_{W}}={\left(K_{\mathrm{PLb}}^{L_{W}}\right)}^{2}$. As a consequence, the binding affinity increases supralinearly with ligand lipophilicity, becoming as high as 10^11^ M^−1^ for the set of parameters considered in this simulation.

For the volume of the lipid phase considered in these simulations (1%), the apparent affinity deviates significantly from the intrinsic affinity for ligand lipophilicities higher than 10^2^. As anticipated from Equation (8), lower (higher) volumes of the lipid phase lead to significant deviations for higher (lower) ligand lipophilicities (see Figure A2 in Appendix B).

It is interesting to note that for a given volume of the lipid phase, the dependence of $K_{\mathrm{eqApp}}^{L_{W}}/K_{\mathrm{eq}}^{L_{W}}$ with ligand lipophilicity is exactly the same for all cases considered (Figure 6A–C). The evaluation of the deviation between the apparent and the intrinsic affinity due to ligand lipophilicity may therefore be performed without knowing the details of the protein specificity and is represented in Figure 7.

The colored slabs in Figure 7 correspond to a given range of $K_{\mathrm{eq}}^{L_{W}}$, and the value that would be obtained for $K_{\mathrm{eqApp}}^{L_{W}}$ is obtained from the interception of horizontal lines with the *y-*axis. The dashed line marks an apparent affinity of 10^6^ M^−1^. This same apparent affinity may correspond to intrinsic affinities as high as 10^10^ M^−1^, for the highest ligand lipophilicity and volume of the lipid bilayer considered (plot C). This figure may be used to estimate the intrinsic binding affinity from values of apparent affinity reported in the literature. For an accurate use of Figure 7, it is necessary to know the volume of the lipid phase and the ligand lipophilicity for the specific ligand and the conditions of the assay. The lipid volumes considered in Figure 7 were chosen to represent typical conditions. The smaller volume considered (α_Lb_ = 0.1%, plot A) corresponds to a phospholipid concentration of ≅1 mg/mL, the highest concentration normally used in in vitro studies. The intermediate volume (α_Lb_ = 1%, plot B) corresponds to the conditions found in blood plasma, and the highest volume (α_Lb_ = 10%, plot C) corresponds to conditions of extremely high lipid concentrations, such as those found in mitochondria due to the extensive folding of the inner mitochondrial membrane.

#### 2.2.2. Specific Examples from Literature

In this section, some specific examples reported in the literature for the binding of ligands to the membrane protein P-gp will be presented. This protein was selected based on its relevance in pharmacology, and because it has been extensively studied. Some representative references are selected to exemplify the impact of the lipid bilayer; no attempt has been made to present an extensive revision.

The binding of ligands to P-gp has been characterized in their native membranes, [15,16,17,84,85,86,87] and with the purified protein solubilized in detergent micelles [17], reconstituted in nanodiscs [17,88,89,90] or proteoliposomes of different lipid composition [17,91,92,93,94,95,96]. The total concentration of protein varies from 2.5 μg/mL (0.025% *w*/*v*) for purified P-gp reconstituted in liposomes to 1 mg/mL (0.1% *w*/*v*) when using native membranes of cells overexpressing P-gp. The latter contain several other proteins, with 1% w/w being estimated for the fraction of P-gp [86]. Therefore, the total concentration of P-gp is always very low, and the value of the apparent affinity usually reported (neglecting the ligand associated with the lipid bilayer and the protein, $K_{\mathrm{eqApp}}^{L_{T}}$) should be similar to $K_{\mathrm{eqApp}}^{L_{W}}$ (see Figure 3B). The ratio of P-gp-to-lipid is usually close to 1% w/w in the native membranes [97] but varies widely in the systems containing purified P-gp. The lower ratio of protein is observed when P-gp is solubilized in detergent micelles (0.1% w/w [17]), and the highest ratio is for P-gp reconstituted in proteoliposomes (10% w/w [92,93]). The fractional volume occupied by the lipid phase varies from 0.005% in some assays performed with proteoliposomes [93] to 1% for P-gp solubilized in detergent or reconstituted in nanodiscs [17,90], with typical values being between 0.05 and 0.1%. Thus, for the case of the P-gp assays, the effect of ligand lipophilicity on the apparent affinity should usually be evaluated from Figure 7A and, in some cases, Figure 7B.

In the selected representative publications, a wide range of ligands have been characterized, leading to apparent binding affinities varying from 10^2^ to 10^8^ M^−1^. Ligand lipophilicity was usually evaluated from their CLogP or from their solubility in aqueous medium, but, in some studies, the partition coefficient to POPC lipid bilayers [16,98] or to the specific lipid composition of the membranes in study [15,89,93,95] is reported. Ligand lipophilicity varied by several orders of magnitude, from relatively polar (Log(*K*_P_) from 1 to 3) to very lipophilic (Log(*K*_P_) ≥ 5).

Significant deviations between the intrinsic binding affinity for P-gp and the apparent affinity reported are therefore anticipated, especially for the more lipophilic ligands. Some specific situations where the interpretation of the results may be strongly biased are discussed below.

The reported apparent binding affinity of Verapamil varies from 2 × 10^6^ M^−1^ (*K*_M_ = 0.5 μM) for P-gp in native membranes [87] to 2.9 × 10^4^ M^−1^ (*K*_M_ = 35 μM) for P-gp solubilized in Octyl glucoside micelles [90] and 2.4 × 10^3^ M^−1^ (*K*_M_ = 422 μM) in n-dodecyl β-D-maltoside [17]. Interestingly, the volume of the lipidic phase is 0.01% in the experiments performed with the native membranes but increases to ≅1% in the micelles. A lower apparent affinity for P-gp solubilized in micelles has also been observed for the P-gp inhibitors Tariquidar, Elacridar, and Zosuquidar [17]. In this case, in the presence of the micelles (*V*_Lb_ ≅ 1%), activation of P-gp’s ATPase activity is observed at low concentrations of the ligands, and for Elacridar, inhibition is observed at high concentration (25 μM). In contrast, in the native membranes (*V*_Lb_ ≅ 0.01%), inhibition is observed at sub μM concentrations, suggesting that the amount of modulator bound to P-gp is always very high. These results were taken as evidence of P-gp being functionally different when solubilized in detergent micelles. However, it may simply result from the more extensive association of the ligand to the detergent micelles due to the much higher volume of the lipidic phase, leading to less ligand available to interact with P-gp. Verapamil lipophilicity is moderate (CLogP = 5), but CLogD_pH=6.8_ = 2.6 [99], and Log(*K*_P_) = 2.3 for EggPC liposomes [89]. From Figure 7, one predicts that 1% volume of the lipid phase (plot B) would lead to a decrease of 1 to 2 orders of magnitude in the apparent binding affinity when compared with the intrinsic affinity. This by itself could at least partially explain the decrease observed in the apparent affinity when the assay is performed with P-gp solubilized in detergent micelles. The effects observed for the P-gp inhibitors are even more dramatic, and this agrees with their higher lipophilicity (CLogP = 6.8 and ClogD_pH=6.8_ = 5.2 for Elacridar). From Equation (8), one predicts almost 5 orders of magnitude decrease in the apparent affinity in the presence of the detergent micelles (α_Lb_ = 0.01) but less than 3 orders of magnitude in the assays with native membranes (α_Lb_ = 10^−4^). Thus, if the intrinsic affinity of the inhibitors for P-gp is the same in both systems, a two orders of magnitude lower apparent affinity would be obtained in the presence of the micelles.

Other examples where the ligand association with the lipidic phase may be at least partially responsible for the variations observed in the apparent affinity are when the lipid composition and/or phase of the lipid bilayer is changed. The work by Sharom and coworkers [95] presents a detailed study on the effect of the lipid phase on the apparent affinity for P-gp reconstituted in liposomes. In general, a decrease in the binding affinity was observed when the membrane phase changed from gel to liquid-disorder. The effect of the phase transition on the ligand partition coefficient to the lipid bilayer was also characterized, and an increase was observed for the liquid-disordered phase. Thus, the decrease in the apparent affinity for P-gp reflects the more extensive association of the ligands with the lipid phase. The effect of cholesterol on the binding of several ligands to P-gp, and on the protein’s activity as transporter, have also been characterized [89,93]. The effects observed were dependent on the ligand and on the cholesterol fraction. This complex behavior is in part explained by the effect of cholesterol on the partition coefficient of the ligands to the lipid bilayers, which depends on the fraction of cholesterol and on ligand’s properties [20,21,26,89,93,100,101,102,103].

#### 2.2.3. How Is Protein Specificity Reflected in the Binding Affinity?

From the above analysis, it is clear that the apparent binding affinity cannot be used to obtain information regarding the protein specificity. It will be briefly discussed in this section whether this may be obtained (and how) from the intrinsic binding affinities for the case of membrane proteins.

The equilibrium between the ligand, the aqueous medium and the protein, $K_{\mathrm{eq}}^{L_{W}}$, reflects the differences in chemical potential of the ligand in the two environments (aqueous medium and protein). However, an increase in $K_{\mathrm{eq}}^{L_{W}}$ may simply reflect an increase in the ligand hydrophobicity. In this case, the increase in the binding constant will lead to a more extensive binding in systems with purified protein. However, this non-specific driving force will increase binding to any other non-polar environments, providing little or no enrichment in the protein of interest when in complex media. This problem in the context of water-soluble proteins has been thoroughly discussed and will not be further addressed here; see, e.g., [43,104,105,106,107]. When referring to membrane-proteins, a relevant aspect is to compare the increase in $K_{\mathrm{eq}}^{L_{W}}$ with the variation observed in the ligand lipophilicity $K_{\mathrm{PLb}}^{L_{W}}$. If a similar variation is observed in both parameters, the increase in $K_{\mathrm{eq}}^{L_{W}}$ is most likely due to a variation on the ligand hydrophobicity. In this case, it may be concluded that the interactions established by the ligand with the lipid bilayer and with the protein binding site are similar, and $K_{\mathrm{PP}}^{L_{\mathrm{Lb}}}$ will most likely be unchanged. If $K_{\mathrm{eq}}^{L_{W}}$ increases but $K_{\mathrm{PLb}}^{L_{W}}$ is unchanged or increases to a lower extent, the relative affinity between the lipid bilayer and the protein increases. This outcome most likely reflects specific interactions established between the ligand and the protein.

Therefore, to evaluate protein specificity, it is necessary to know the parameters for all three equilibria: from the aqueous medium towards the protein and the lipid bilayer, and from the lipid bilayer towards the protein. We performed this recently using the interaction of a homologous series of amphiphiles with the membrane-protein P-glycoprotein [15]. The polar portion of the amphiphiles was maintained, and the length of their alkyl chain was increased from 4 to 8 carbons, leading to an increase in their hydrophobicity. As expected, the lipophilicity of the amphiphiles increased with the length of the alkyl chain [108]. The apparent binding affinity towards P-glycoprotein when calculated considering the total ligand concentration ($K_{\mathrm{eq}}^{L_{T}}$) also increased, suggesting an increase in the protein affinity. However, when the results were analyzed with the adequate formalism, it was observed that $K_{\mathrm{eq}}^{L_{W}}$ was almost unchanged, and that $K_{\mathrm{PLb}}^{L_{\mathrm{Lb}}}$ decreased with the increase in the length of the alkyl chain. This allowed us to conclude that this membrane protein does not interact specifically with groups with very low polarity, such as alkyl chains.

## 3. Discussion

The simulations shown in this manuscript show that the presence of the lipid phase cannot be neglected when characterizing the interaction of ligands with proteins in complex systems. This is the case for both water-soluble and membrane proteins, and for hydrophilic or lipophilic ligands. It is shown that ligand sequestration in the lipid bilayer always leads to a decrease in the apparent binding affinity, irrespective of the protein being water-soluble or a membrane protein, with the binding pocket accessible from the aqueous medium or from the membrane. The presence of the lipid bilayer only increases the apparent affinity in the case of very polar ligands. In this case, when the partition coefficient between the aqueous medium and the lipid membrane is lower than 1, the exclusion of the ligand from the lipid bilayer increases the concentration in the aqueous medium and in association with the protein.

The presented formalism allows obtaining the intrinsic binding affinity, and it must be used when the goal is to evaluate protein specificity. Not doing so leads to apparent affinities that do not reflect the strength of the interactions between the ligand and the protein. Furthermore, the obtained apparent affinity depends on the system properties, precluding the quantitative comparison between data obtained under different conditions. This is particularly misleading when the apparent affinity is obtained by considering the total ligand concentration, i.e., neglecting the ligand associated with both the lipid bilayer and the protein.

The procedure to obtain quantitative information on the intrinsic affinity of ligands to proteins in the presence of lipid bilayers must consider: (i) the volume of the lipid bilayer; (ii) the partition coefficient of the ligand between the aqueous phase and the lipid bilayer; and (iii) the use of a formalism that explicitly includes the distribution of the ligand between the distinct media (aqueous, lipid bilayer, and protein). If the protein has a single binding site to which only one ligand molecule can bind, the system of equations for the equilibrium distribution of the ligand has a closed-form analytical solution, Equations (4) and (5), and the intrinsic binding affinity is easily obtained. If, on the other hand, the protein contains several binding sites, or if the binding site can accommodate several ligand molecules, the equation that allows calculating the concentration of free ligand in the aqueous medium cannot be solved analytically and must be obtained numerically [43,44,45,46,47,48,49,50,51]. The formalism explored in this work has recently been applied for the case of ligand binding to the membrane protein P-glycoprotein [15], whose large binding pocket can bind several ligand molecules [77,93,109,110,111,112,113]. In this case, the binding of several ligand molecules had to be considered, and the concentration of free ligand was obtained numerically. More importantly, although the partition coefficients for the three equilibria are still related by Equation (2), the relation between the partition coefficients towards the protein and the corresponding binding constants depends on the number of ligand molecules that can bind to the protein [1,15]. In spite of the higher complexity of the system, it was still tractable using the software Excel^®^ (Microsoft Office Professional Plus 2019, Microsoft, Redmond, WN, USA), a tool that is widely available in the scientific community. The intrinsic binding affinities thus obtained allowed the quantitative comparison between different assays and for several ligands, providing important insight regarding protein specificity.

It is anticipated with great expectation that the generalized use by the scientific community of formalisms such as the one described in this manuscript will allow the clarification of ligand specificity for membrane proteins. This will have an enormous impact on the development of more efficient and more specific drugs targeted for membrane proteins.

## Figures and Tables

**Figure 1 molecules-28-03136-f001:**
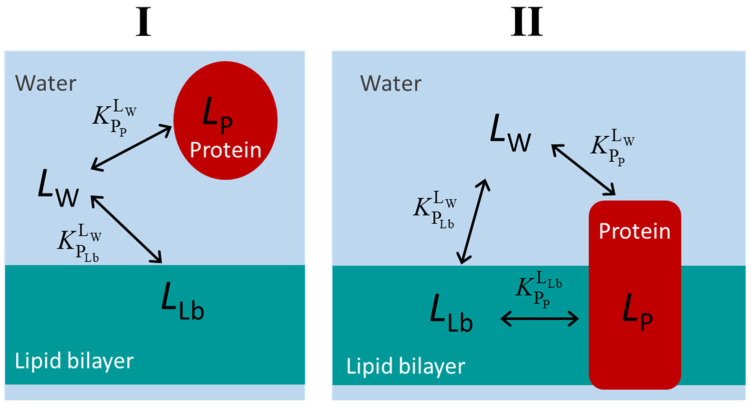
Kinetic schemes for the equilibrium distribution of the ligand between the aqueous medium, a protein soluble in the aqueous medium (case (**I**)) or in the lipid membrane (case (**II**)), and the membrane lipid bilayer. The association of the ligands to the protein is considered to occur to a single and well-defined binding site (saturable binding), while a partition (non-saturable) is considered for the association with the lipid bilayer.

**Figure 2 molecules-28-03136-f002:**
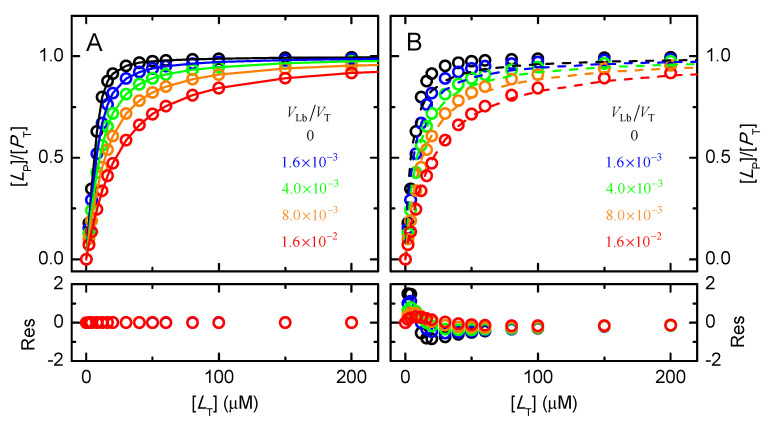
Dependence of the protein saturation with ligand ([*L*_P_]/[*P*_T_]) with the total ligand concentration, for a protein soluble in the aqueous phase with M_W_ = 50 kDa and a total concentration [*P*_T_] =10 μM (corresponding to *V*_P_ = 4.2 × 10^−4^ *V*_T_, $K_{\mathrm{eq}}^{L_{W}}$ = 10^6^ M^−1^, and $K_{\mathrm{PLb}}^{L_{W}}$ = 10^3^; in the absence of a lipid bilayer and in the presence of increasing lipid concentrations as indicated in the plots. The lines are the best fits neglecting the sequestration of the ligand by the lipid bilayer, considering either the concentration of ligand in the aqueous medium, Equation (6) (plot (**A**)) or the total ligand concentration, Equation (7) (plot (**B**)). The residuals of the best fit are shown in the bottom plots.

**Figure 3 molecules-28-03136-f003:**
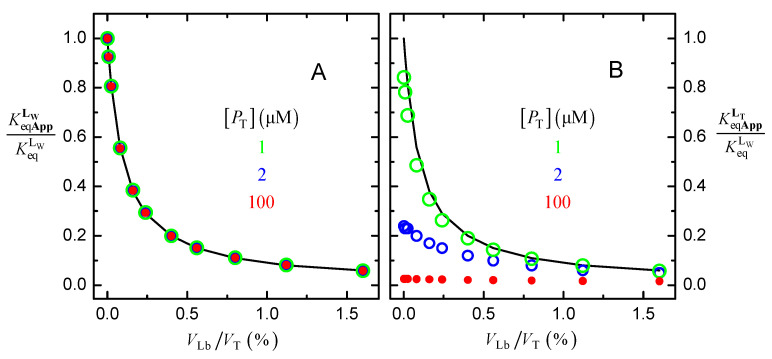
Dependence of the apparent binding affinity on the fractional volume of the lipid bilayer; when considering that the ligand is either in the aqueous medium or associated with the protein ($K_{\mathrm{eqApp}}^{L_{W}}$ Equation (6), plot (**A**), and when assuming large excess of ligand ($K_{\mathrm{eqApp}}^{L_{T}}$ Equation (7), plot (**B**). The parameters considered in the simulations were: $K_{\mathrm{eq}}^{L_{W}}$ = 10^6^ M^−1^, and $K_{\mathrm{PLb}}^{L_{W}}$ = 10^3^, for different protein concentrations, as indicated in the plots. The line is the prediction from Equation (8).

**Figure 4 molecules-28-03136-f004:**
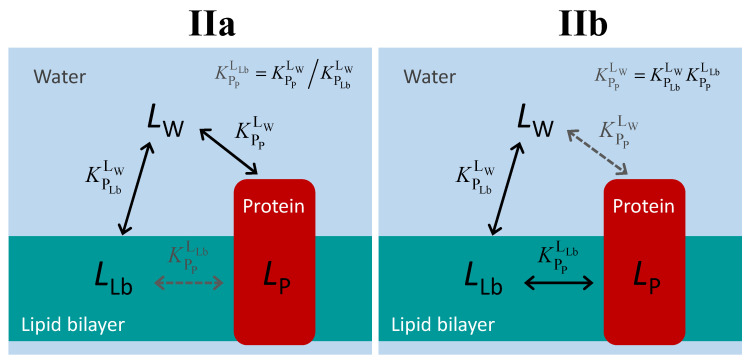
Schematic representation of the ligand equilibrium distribution between the lipid bilayer and a membrane protein for the case of ligand-protein binding from the aqueous medium (**IIa**) or from the lipid bilayer (**IIb**). The equilibria indicated in black correspond to paths that may be followed by the ligand, while the equilibrium in grey results only from the impositions of the thermodynamic cycle.

**Figure 5 molecules-28-03136-f005:**
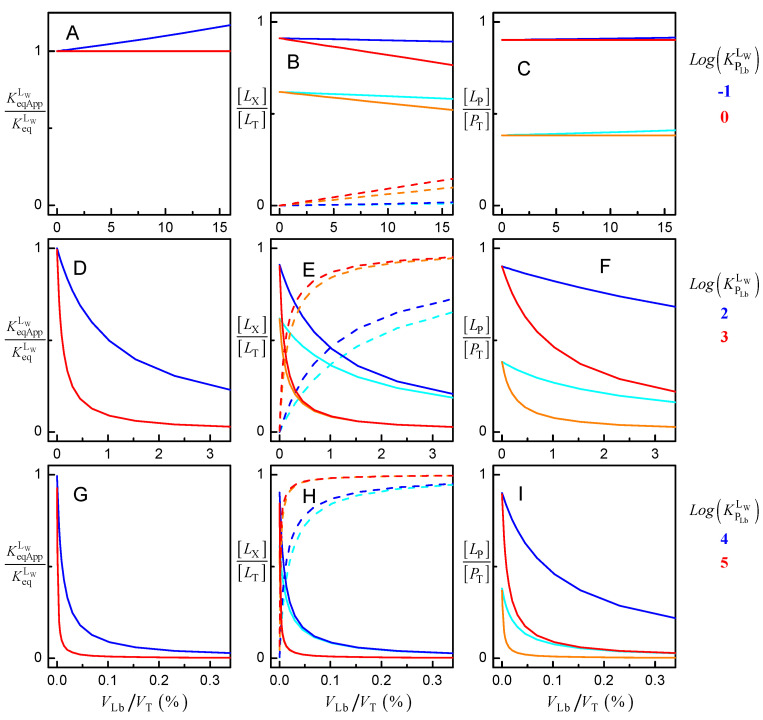
Effect of the volume of the lipid bilayer on the equilibrium association of ligand with a protein with M_W_ = 50 kDa, P_T_ = 1 μM and $K_{\mathrm{eq}}^{L_{W}}$ = 10^6^ M^−1^ for different ligand lipophilicity, as indicated in the figure. The variation of the apparent binding affinity is shown in the left plots (**A**,**D**,**G**), the fraction of ligand in the aqueous medium (continuous lines) and in the lipid bilayer (dashed lines) is shown in the middle plots (**B**,**E**,**H**), and the protein saturation with ligand is shown in the right plots (**C**,**F**,**I**); for [*L*_T_] = 1 μM (light colors) and 10 μM (dark colors).

**Figure 6 molecules-28-03136-f006:**
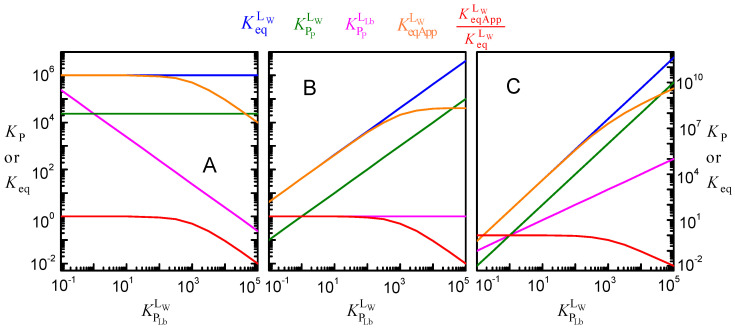
Effect of ligand lipophilicity on the equilibrium constants for the distribution between the aqueous phase, the lipid bilayer, and the membrane protein, assuming that the intrinsic affinity of the ligand to the protein is unchanged ($K_{\mathrm{eq}}^{L_{W}}$ = 10^6^ M^−1^, plot (**A**)), that the relative affinity for the lipid bilayer and for the protein is unchanged ($K_{P_{P}}^{L_{\mathrm{Lb}}}$ = 1, plot (**B**)), or that the affinity for the protein increases supralinearly with ligand lipophilicity ${\left(K_{P_{P}}^{\mathrm{LLw}}=\left(K_{\mathrm{PLb}}^{\mathrm{LLw}}\right)\right)}^{2}$ = 1, plot (**C**)); for *P*_T_ = 1 μM, *M*_W_ = 50 kDa, and *V*_Lb_ = 0.1%).

**Figure 7 molecules-28-03136-f007:**
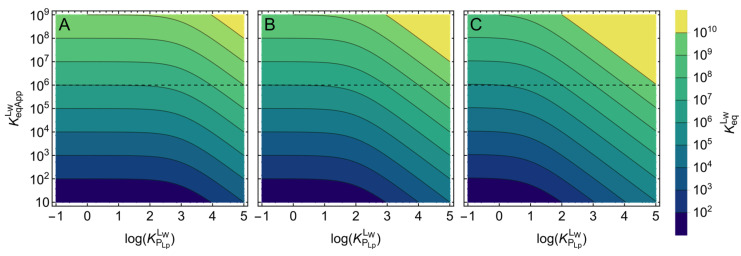
Effect of ligand lipophilicity ($K_{\mathrm{PLb}}^{L_{W}}$) on the apparent affinity ($K_{\mathrm{eqApp}}^{L_{W}}$) for different values of the intrinsic affinity ($K_{\mathrm{eq}}^{L_{W}}$) of binding to a protein in the presence of a lipid bilayer. The colored slabs correspond to a given value of $K_{\mathrm{eq}}^{L_{W}}$ (see scale on the right), the value that would be obtained for $K_{\mathrm{eqApp}}^{L_{W}}$ is the intercept of a horizontal line with the *y*-axis (exemplified by the dashed line at $K_{\mathrm{eqApp}}^{L_{W}}=10^{6}$). The volume occupied by the protein was considered negligible, and the volume of the lipid phase was 0.1% (plot (**A**)), 1% (plot (**B**)), and 10% (plot (**C**)).

## Data Availability

Not applicable.

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
