# Peer review of "Ligand’s Partition to the Lipid Bilayer Should Be Accounted for When Estimating Their Affinity to Proteins"

_molecules, 2023, doi:10.3390/molecules28073136_

Round 1

Reviewer 1 Report

Moreno and Salvador's manuscript addresses an important and, as the authors note, almost always overlooked point. Namely, how the binding of a ligand with a protein target is the result of competition between different binding sites. In the manuscript, the authors make an equilibrium K correction by considering partitioning into lipid membranes.

The work is well structured and the conclusions shareable. Therefore, the work is certainly worthy of publication. 

I suggest, however, that the authors better define the terms of Equation 1. Reading the paper one understands who KPp is, but it is the authors' job to make the reader's job easier. Figures 1 and 4 could also be improved.

Author Response

Moreno and Salvador's manuscript addresses an important and, as the authors note, almost always overlooked point. Namely, how the binding of a ligand with a protein target is the result of competition between different binding sites. In the manuscript, the authors make an equilibrium K correction by considering partitioning into lipid membranes.

The work is well structured and the conclusions shareable. Therefore, the work is certainly worthy of publication. 

The authors thank the reviewer for the critical reading of the manuscript and for the positive evaluation of the work.

I suggest, however, that the authors better define the terms of Equation 1. Reading the paper one understands who KPp is, but it is the authors' job to make the reader's job easier. Figures 1 and 4 could also be improved.

KPp has been defined (lines 142-143). We could not however understand what where the reviewer’s specific concerns regarding Figures 1 and 4. Therefore we have done only small alterations, namely on the position of the arrows and equilibrium constants to improve the symmetry in the kinetic scheme (Figure 1 and 4), and in Figure 4 we added the relation between the equilibrium constant that results from the thermodynamic cycle (in grey), and those that correspond to paths that may be followed by the ligand (in black). 

Reviewer 2 Report

Comments on molecules-2280444:

The current paper describes a mathematical consideration of the ligands’ partition between different phases (water, protein and lipid), with some parameter scans, formula derivation and brief discussions. The topic fits the ‘molecules’ scope and could be considered for publication. Below are some detailed comments.

For simplicity, the authors assume that only a single binding site exists. Although this avoids the extra complexity due to multi-site binding, I would point out that there are many other potentially influencing factors that enter the equation and should be clarified. For example, probably the most general factor is the protonation-state variation, which is determined by the pH condition, the pKa of ligands/drugs, and the perturbation caused by environments (pKa shifts compared with aqueous solution). The protonation-deprotonation equilibria are rather common in drug applications. E.g., the abused drug Ketamine has a pKa value (~7.5) very close to the physiological condition pH 7.4. In the modelling of binding between such drugs and proteins or macrocyclic hosts, often both protonation states are modelled and averaged to derive the apparent binding thermodynamics.  

The presentation of Figure 2, 3, 5 and A1 could be bettered by explicitly adding a legend, instead of explaining the coloring regime in the caption.

Playing with purely mathematical models and doing parameter scans are interesting. For example, in Figure 6, pronounced variations are observed for K_app/K. However, it would be beneficial to substitute the parameters with values really measured experimentally, in order to provide some insights into the impact of the authors’ results (partition between different phases/media) in practical systems. After reading the paper, a straightforward question for the readers to ask is when such partition considerations are necessary. I.e., neglecting these phenomena would introduce significant systematic biases.  

Author Response

The current paper describes a mathematical consideration of the ligands’ partition between different phases (water, protein and lipid), with some parameter scans, formula derivation and brief discussions. The topic fits the ‘molecules’ scope and could be considered for publication. Below are some detailed comments.

The authors thank the reviewer for the critical reading of the manuscript and for the suggestions.

For simplicity, the authors assume that only a single binding site exists. Although this avoids the extra complexity due to multi-site binding, I would point out that there are many other potentially influencing factors that enter the equation and should be clarified. For example, probably the most general factor is the protonation-state variation, which is determined by the pH condition, the pKa of ligands/drugs, and the perturbation caused by environments (pKa shifts compared with aqueous solution). The protonation-deprotonation equilibria are rather common in drug applications. E.g., the abused drug Ketamine has a pKa value (~7.5) very close to the physiological condition pH 7.4. In the modelling of binding between such drugs and proteins or macrocyclic hosts, often both protonation states are modelled and averaged to derive the apparent binding thermodynamics.  

We thank the reviewer for calling attention to this important aspect. Throughout the manuscript we were focused on the comparison between ligands with distinct lipophilicity. However, the effect of the lipid bilayer is relevant when the affinity of the ligand for the membrane changes. This being because the ligand is different, or because the solvation properties of the membrane are changed. This may be due to changes in ligand’s ionization (as pointed out by the reviewer) but also by changes in the lipid composition of the membrane. Those aspects have now been discussed in the revised manuscript. New text was inserted in the introduction (lines 71-84 in the “ChangesMarked” version of the manuscript), and a disclaimer was introduced in lines 368-372 regarding the meaning of a distinct ligand lipophilicity being due to changes in the lipid bilayer properties or in the ligand ionization state.

The presentation of Figure 2, 3, 5 and A1 could be bettered by explicitly adding a legend, instead of explaining the coloring regime in the caption.

The figures have been changed to include the identification of the different colors. We thank the reviewer for this suggestion that will significantly facilitate the interpretation of the figures. 

Playing with purely mathematical models and doing parameter scans are interesting. For example, in Figure 6, pronounced variations are observed for K_app/K. However, it would be beneficial to substitute the parameters with values really measured experimentally, in order to provide some insights into the impact of the authors’ results (partition between different phases/media) in practical systems. After reading the paper, a straightforward question for the readers to ask is when such partition considerations are necessary. I.e., neglecting these phenomena would introduce significant systematic biases.  

To answer this reviewer concern we have added a new section to the manuscript (section 2.2.2, lines 495-571). Two approaches were followed: i) some specific examples for typical values of ligand lipophilicity and apparent affinities reported in literature are presented, indicating the estimated bias introduced by the analysis using the apparent affinity; and ii) the effects of changing the composition and phase of the lipid bilayer are discussed.

Round 2

Reviewer 2 Report

The paper as a good addition to literatures is acceptable in the current form.